# A Simple Optical Sensor Based on Multimodal Interference Superimposed on Additive Manufacturing for Diameter Measurement [note 1]

**DOI:** 10.3390/s22124560

**Published:** 2022-06-17

**Authors:** Victor H. R. Cardoso, Paulo Caldas, Maria Thereza R. Giraldi, Cindy Stella Fernandes, Orlando Frazão, João C. W. Albuquerque Costa, José Luís Santos

**Affiliations:** 1Applied Electromagnetism Laboratory, Federal University of Pará, Rua Augusto Corrêa, 01, Belém 66075-110, Brazil; joao.weyl@gmail.com; 2Institute for Systems and Computer Engineering, Technology and Science, Rua do Campo Alegre, 687, 4169-007 Porto, Portugal; orlando.frazao@inesctec.pt (O.F.); josantos@fc.up.pt (J.L.S.); 3Polytechnic Institute of Viana do Castelo, Rua Escola Industrial e Comercial de Nun’Álvares, 4900-347 Viana do Castelo, Portugal; 4Laboratory of Photonics, Military Institute of Engineering, Praça Gen. Tibúrcio, 80, Rio de Janeiro 22290-270, Brazil; mtmrocco@gmail.com; 5Faculty of Computing and Electrical Engineering, Federal University of South and Southeast of Pará, Marabá 68507-590, Brazil; cindy.fernandes@gmail.com; 6Department of Physics and Astronomy, Faculty of Sciences of University of Porto, Rua do Campo Alegre, 687, 4169-007 Porto, Portugal

**Keywords:** SMS, optical strain gauge, optical sensor, diameter monitoring

## Abstract

In many areas, the analysis of a cylindrical structure is necessary, and a form to analyze it is by evaluating the diameter changes. Some areas can be cited: pipelines for oil or gas distribution and radial growth of trees whose diameter changes are directly related to irrigation and the radial expansion since it depends on the water soil deficit. For some species, these radial variations can change in around 5 mm. This paper proposes and experimentally investigates a sensor based on a core diameter mismatch technique for diameter changes measurement. The sensor structure is a combination of a cylindrical piece developed using a 3D printer and a Mach–Zehnder interferometer. The pieces were developed to assist in monitoring the diameter variation. It is formed by splicing an uncoated short section of MMF (Multimode Fiber) between two standard SMFs (Singlemode Fibers) called SMF-MMF-SMF (SMS), where the MMF length is 15 mm. The work is divided into two main parts. Firstly, the sensor was fixed at two points on the first developed piece, and the diameter reduction caused dips or peaks shift of the transmittance spectrum due to curvature and strain influence. The fixation point (FP) distances used are: 5 mm, 10 mm, and 15 mm. Finally, the setup with the best sensitivity was chosen, from first results, to develop another test with an optimization. This optimization is performed in the printed piece where two supports are created so that only the strain influences the sensor. The results showed good sensitivity, reasonable dynamic range, and easy setup reproduction. Therefore, the sensor could be used for diameter variation measurement for proposed applications.

## 1. Introduction

In many areas, diameter change control is observed and plays an important role. In the first example, the oil or gas transportation pipelines deform because they present a high pressure, causing fatigue, deformation, and disruption [1]. Additionally, deformation can be caused by natural disasters such as landslides and earthquakes or the theft of material transported in pipelines. The corrosion process is also one of the main factors contributing to accidents in pipelines [2]. In these cases, their disruption can produce substantial environmental injury, product failure, monetary problems, and occasionally the loss of life from explosions [3,4]. Another application is the monitoring of the variation of tree stem diameter. This parameter is directly related to irrigation due to the dependence on the soil water deficit [5,6]. Additionally, tree growth is affected by the CO${}_{2}$ rate and air pollutants and is essential in the global circulation of heat and water [7,8,9]. The use of optical fiber sensors (OFS) for these cases is interesting because, due to characteristics such as immunity to electromagnetic interference, compact size, resistance to extreme ambients, low attenuation, lightweight, survival in harsh environments, and the possibility of transmission over long distances.

Multimodal Interference (MMI) devices based on optical fibers have been extensively investigated and act as sensors in recent years for different applications, such as temperature [10], strain [11,12,13], air pressure [14], vibration [15], gasohol quality [16], structure health monitoring [17], flow rate [18,19], and so on. There exist several types of MMI structures, such as the Sagnac Interferometer [20], Fabry–Perot Interferometer [21], and Singlemode-Multimode-Singlemode (SMS) [22]. One of the most common MMI structures is the SMS [22,23]. This structure is formed by a short segment of Multimode fiber (MMF) spliced between two single-mode fibers (SMFs). An SMS sensor is based on the mechanism of a mode-coupling to transfer optical power between the core mode and higher-order modes of the cladding through core diameter mismatch. In addition, the SMS presents the advantages of optical fiber over conventional measurement techniques, ease of fabrication, low cost of production, flexible design, compact size, and high sensitivity [24,25,26].

The SMS structure has been widely used in curvature analysis in recent years for strain analysis [22,27,28]. The sensor is fabricated by a simple technique of fusion splicing of a small region of MMF with two SMFs. Other sensor structures, like FBG or LPG, usually require special optical fibers, high-cost micro machining instruments, complex fabrication processes, or/and expensive coating materials. Furthermore, according to the literature [11,12,29,30], when the SMS sensor is submitted to axial strain, it causes a shift in the multimode interference spectrum due to optical fiber dimensions change. Liu et al. [30] demonstrated a static strain fiber ring cavity laser sensor based on an SMS structure. The axial strain is applied in the sensor, and dimensions change results in shifts of the central wavelength. The high sensitivity of 2.3 pm/μϵ is obtained, and good linearity of the sensing system can be observed; it is possible to be used in a long-distance axial strain sensing system. In Sun et al. [31], an optical strain sensor utilizing the twist effect of multimode fiber is proposed. The MMF section is submitted to the heating process, which is then twisted. The experimental results indicated a high strain sensitivity of 42.5 pm/μϵ. Hatta et al. [29] show a strain measurement technique using a single mode-multimode-single mode fiber structure sensor and an optical time-domain reflectometer. A strain measurement range of 0–1000 μϵ can be performed with a resolution of better than 10 μϵ.

Additive manufacturing (AM), popularly known as a 3D printer, already plays a crucial role in Industry 4.0 for manufacturing industries, with broad applications including high-tech industries, such as aerodynamic, biomedical, aerospace, and civil engineering. AM has several advantages: scrap reduction, design versatility, automation possibilities, development of lighter and more robust products, in addition to hardness, chemical and impact resistance, transparency, and ductility [32]. There are many substrates of AM, such as ABS material, polymeric materials (PET-G material, polyamide material, polylactic acid (PLA) material, and so on [17,32,33,34]. Among the most used polymers, we can mention PLA. This material was selected as the printing material in this work.

AM also has been used in recent years with optical fiber sensors to optimize the measurement [35,36]. Hong et al. [37,38] developed and fabricated a fiber Bragg grating pressure sensor using a 3D printing method for the measurement of vertical pressure using PLA material. Another example of OFS embedding within structural materials based on AM is used for fatigue [39], strain [40] and deformation [41], curvature [28], and evaluating the integrity of the structures [42].

In this work, we investigate experimentally a method for diameter monitoring using a traditional SMS structure combined with a piece developed using a 3D printer. We use an experimental setup based on an SMS fixed on a structure developed in a 3D printer. Thus, it was possible to perform the analysis of the diameter variation of different structures. The SMS sensor is put on the strain gauge and fixed on it at two points where the distance of fixation points is 5, 10, and 15 mm.

Section 2 presents the theoretical principle of the SMS submitted to strain, including the interface coupling factors and effects on the output power profile are discussed. Section 3 presents the simulation analysis using the finite element method, the experimental setup, and the sensing mechanism. The results and discussion are described in Section 4, and, at the end of this section, the comparison between the results is shown. In Section 5, the conclusions are exposed due to the utilization of the proposed sensor as an MMI sensor for diameter measurement.

## 2. Theoretical Principle of the SMS Sensor Submitted to Strain

A traditional SMS structure consists of a short MMF section spliced between two SMFs, as depicted in Figure 1. Due to the large core diameter mismatch, the MMF section of the proposed sensor works as a core-cladding coupling mechanism. The light is injected from the input SMF into the MMF, where the fundamental and higher order modes are excited, and interference between the different modes occurs while the light beam propagates along the MMF section [15,19]. Due to higher modes excitation, the optical power coupled to the output SMF depends on the amplitudes and phases of the MMF modes, and this dictates the spectral transmission response at the output SMF [43,44,45,46]. The MMF couples part of the light that travels along with the core of the input SMF to the cladding of the output SMF, and this coupling induces a loss of power in the transmitted signal that travels along the core [43,47,48]. If we consider that the MMF and SMF sections are perfectly axially aligned, the optical transmission power (*P*) can be given as [43,47,48]:(1)$$P={\left|a_{0}^{2}+a_{1}^{2}Exp\;i\left(\beta_{0}-\beta_{1}\right)L_{MMF}+a_{2}^{2}Exp\;i\left(\beta_{0}-\beta_{2}\right)L_{MMF}+\cdots +a_{n}^{2}Exp\;i\left(\beta_{0}-\beta_{n}\right)L_{MMF}\right|}^{2},$$
where $a_{n}$ are the field amplitudes of the *n* modes at the first interface (I_1_), $\beta_{n}$ are the propagation constants of the *n* modes, and *L_MMF_* is the length of the MMF. It is evident that *P* is influenced by strain applied to the MMF with some change in $\beta_{n}$ and *L_MMF_*.

It is known that the power of the MMF $LP_{0m}$ mode is defined from the coupling coefficient from the SMF $LP_{01}$ mode to the fundamental mode in the MMF. In the MMF and output SMF interface (I_2_), the phase difference between two modes, $LP_{0m}$ and $LP_{0n}$, has the widest excitation coefficient. It is determined by the L_MMF_ and the difference of the longitudinal propagation constants between two radial modes, $\beta_{m}-\beta_{n}$, and it is given by:(2)$$\beta_{m}-\beta_{n}=\frac{u_{m}^{2}-u_{n}^{2}}{2ka^{2}n_{core}},$$
where $u_{m}^{2}=\pi \left(m-1/4\right)$ and $u_{n}^{2}=\pi \left(n-1/4\right)$ are the roots of the zeroth-order Bessel function, *a* is the MMF core radius, k=2π/λ is the free-space wave number, λ is the wavelengths, and $n_{core}$ is the refractive index of the fiber core.

Interference among different modes happens at the exit end of the MMF. The condition for an interference pattern between $LP_{0m}$ and $LP_{0n}$ modes is:(3)$$\Delta \phi =\left(\beta_{m}-\beta_{n}\right)L_{MMF}=2\pi N,$$
where *L_MMF_* is the propagation distance and *N* is an integer, respectively.

From Equations (Equation 2) and (Equation 3), the wavelength for a constructive or a destructive interference, according to the MMI theory, can be expressed as:(4)$$\lambda =\frac{8a^{2}n_{core}N}{\left(m-n\right)\left[2\left(m+n\right)-1\right]L_{MMF}}$$
where *m* and *n* are the mode numbers.

From the derivation of Equation (Equation 4), we have the refractive index of the core, the diameter of the core, the length and the modes m, n of the MMF permit to determine the wavelength of the SMS sensor [49]. The wavelength shift depends on the change in axial strain, as given by [30]:(5)$$\Delta \lambda =(\frac{\Delta n_{core}}{n_{core}}+\frac{2\Delta a}{a}-\frac{\Delta L_{MMF}}{L_{MMF}})\lambda$$
where *L_MMF_* is the MMF section length, *a* is the core radius, $\Delta L_{MMF}$ is the induced change in the MMF section length, Δa is the induced change in the core radius, and $\Delta n_{core}$ is the induced change in the refractive index. The sensor will have an increase in its dimensions due to the applied strain (ϵ). For an applied strain, the change in the MMF length, core radius, and refractive index can be expressed respectively as [12,30,50]:(6)$$\Delta L_{MMF}=L_{MMF}\times \epsilon$$
(7)Δa=−ν×a×ϵ
(8)$$\Delta n_{core}=\frac{n_{core}^{3}}{2}\left[\rho_{12}-\nu \left(\rho_{11}+\rho_{12}\right)\right]\epsilon$$
(9)$$\Delta n_{core}=-\rho_{e}\epsilon$$
where ν is the Poisson ratio of the fiber, $\rho_{11}$ and $\rho_{12}$ are the Pockel’s coefficients, and $\rho_{e}$ is the effective strain-optic coefficient. Equation (Equation 5) can be rewritten as:(10)$$\Delta \lambda =-(1+2\nu +\frac{\rho_{e}}{n_{core}})\epsilon \times \lambda$$

## 3. Simulation Analysis and Experimental Setup

The theoretical static investigation was carried out to confirm the feasibility of the proposed strategy and structures using a Finite Element Method (FEM). As shown in Figure 2, one side of the sensor was fixed and constrained and at the other side the displacement was applied. The distribution of strain can be observed in Figure 3. Due to the displacement applied on the piece, the mid-region is the part of the piece where the greatest stress occurs. Based on it, we decided to fix the sensing head at two points. In this way, the tension effect associated with the displacement is increased.

The material chosen was PLA. This material is widely utilized in the production of structures fabricated with the aid of 3D printers. It also has a 49–52 ${}^{\circ }$C heat deflection temperature, which means that the structure begins to soften [51]. PLA has a glass transition temperature between 50–80 ${}^{\circ }$C, which is the point at which a material changes state [52]. Considering, for example, that the sensor will be used in the Amazon, where the climate is characterized by being hot and humid, with temperatures between 22–35 ${}^{\circ }$C, and PLA is a good candidate to be used for this monitoring. The physical parameters of the principal elements of the developed sensor are shown in Table 1.

From Figure 3, we can see that the strain is concentrated nearly to the mid-region of the piece. This is due to the displacement generated in the piece that forces the mid-region to suffer greater stress. Therefore, it was first decided to conduct tests with the fixed sensors in the region of greatest stress. Based on the FEM, it was decided to change the structure so that the sensor could be more tensioned and suffer less curvature effect. The SMS sensor is influenced by curvature. The sensor output power and spectrum intensities are functions of the curvature radius. When the bending is changed, the interference pattern is altered [19]. Thus, we created two supports in the mid-region of the piece, as illustrated in Figure 4. Similarly, in the first analysis, we used the FEM investigation to confirm the feasibility of the proposed strategy and structure, as depicted in Figure 4. It is possible to observe that the supports suffer a shift between them, in the maximum stress zone. This means that, when the 3D printer piece is submitted to a diameter modification, the sensor suffers a maximum stress due to the displacement between the supports. Since the sensor is fixed, the compression causes a deformation in the sensor.

A second simulation analysis was performed to demonstrate if the optical fiber would be stressed when placed over the printed piece. Figure 5 showed the axial stress direction that the optical fiber is submitted using the piece with supports. The arrows illustrate that the sensor head is tensioned in opposite directions and, due to this, the sensor presents a change in the wavelength response. It was possible to know that the optical fiber would be tensioned and change its dimensions. Therefore, this axial stress in the sensor will cause its dimensions to change, leading to a variation in the wavelength.

In this work, an experimental setup was proposed as shown in Figure 6. The transmission spectrum of the SMS sensor was monitored with an Optical Spectrum Analyzer (Anritsu, MS9740A, Atsugi, Kanagawa, Japan). The sensors assembled in this work are fabricated with the core/cladding diameters of the SMF 8.2/125 μm (Corning SMF-28, Inc., Corning, NY, USA) and, for the step-index MMF section, the core/cladding diameters are 105/125 μm (FG105-LCA from Thorlabs, Inc., Newton, NJ, USA) It is important to emphasize that, for the length of the MMF section used in this work, it is not possible to observe the self imaging phenomenon because the length of the MMF is much shorter than the length needed to attain this phenomenon. According to Soldano [49], the self imaging phenomenon can be described as the property of multimode waveguides where an input field profile is produced in single or multiple images at periodic intervals along the propagation direction of the guide.

In this work, two different pieces were used based on the simulation analysis—one of them with supports and the other without. The linear translation stage was used to apply the displacement, that is, diameter variation, and, due to this, the sensor is tensioned. In the piece without supports, the sensing head is fixed on the piece at the mid-region, which is the greatest stress region. Two pieces without supports were developed with different diameters (D_S_), 80 mm and 110 mm, to assist in monitoring the diameter changes (Δd). The sensing head was fixed at two points on the pieces, with different distances between them. Three values were used for the spacing between the fixation points (FP) equal to 5 mm, 10 mm, and 15 mm. When the displacement occurs, the optical fiber is tensioned and bent because the sensors were fixed on the piece, as illustrated in Figure 7a. The red dashed line represents the final position of the piece after the displacement is applied. The sensors present dips or peaks shift of the transmittance spectrum due to the induced curvature and strain.

In the second case, the piece presents two supports placed in its mid-region. The piece diameter (D_S_) is equal to 80 mm. The mechanical operation of the piece with supports is depicted in Figure 7b. The sensors are fixed on the supports, and, when the displacement occurs, the optical fiber is stressed due to the enhancement in the gap between the two supports. The parameters of the 3D printer process used were the extrusion temperature and the printing speed of the 200 ${}^{\circ }$C and 30 nm/s, respectively, as observed in the literature [53].

## 4. Results and Discussion

### 4.1. Without Supports

Firstly, the SMS sensor was placed on the cylindrical structure with D_S_ = 80 mm, and it was fixed at 5, 10, and 15 mm of the total length of the MMF section. Figure 8 shows the recorded spectra of this experiment, and Figure 9 shows the second-order polynomial fitting result of the corresponding wavelength shift as a function of the displacement of the diameter piece for the three FP values. The sensor sensitivity was obtained from the slope of the first derivative of the second-order polynomial curve.

Figure 8a shows the dip shift of the transmittance spectrum due to the induced curvature and strain in FP = 5 mm. It was possible to obtain a sensitivity of —0.876 nm/mm, coefficient of determination ($R^{2}$) of 0.9909 and the dynamic range (Δd) of 5 mm, as depicted in Figure 9a. After that, the FP was increased to 10 mm. Figure 8b shows the dip shift of the transmittance spectrum due to the induced curvature strain with FP = 10 mm. The $R^{2}$ showed a small increase to 0.9954. The sensitivity obtained for this configuration was —0.3892 nm/mm and the dynamic range of 4 mm is observed in Figure 9b. Figure 8c shows the dip shift of the transmittance spectrum due to the induced curvature strain with FP = 15 mm. It was possible to observe an increase in the sensitivity to —0.768 nm/mm when comparing with the result for FP = 10 mm. The dynamic range showed a reduction to 2 mm, as depicted in Figure 9c. The polynomial behaviour was observed and presents a $R^{2}$ of 0.9811.

The second part of the experiment is similar to what was performed previously, but with D_S_ = 110 mm. Figure 10 shows the recorded spectra of this experiment and Figure 11 shows the second-order polynomial fitting result of the corresponding wavelength shift as a function of displacement of the diameter piece for the three FP values.

For FP = 5 mm, Figure 10a shows the peak shift of the transmittance spectrum due to the induced curvature strain; it was possible to obtain a sensitivity of —0.22 nm/mm, $R^{2}$ of 0.9979, and a dynamic range of 8 mm, as depicted in Figure 11a. After that, the FP was increased to 10 mm. Figure 10b shows the peak shift of the transmittance spectrum due to the induced curvature strain for FP = 10 mm.

The $R^{2}$ showed a decrease to 0.9888. The sensitivity obtained for this configuration was —0.2284 nm/mm and the dynamic range of 3 mm as observed in Figure 11b. When FP = L_MMF_ = 15 mm, as illustrated in Figure 10c, the peak shift of the transmittance spectrum due to the induced curvature and strain is presented. It was possible to observe an increase in the sensitivity to —0.691 nm/mm when comparing it with the result for FP = 10 mm. The $R^{2}$ and dynamic range showed a reduction, respectively, to 0.9892 and 3.5 mm. The polynomial behavior can be observed in Figure 11c.

### 4.2. With Support

Based on the results presented previously and FEM analysis, it was decided to modify the piece. The piece was redesigned so that the sensor was subjected only to stress. The parameters selected were D_S_ equal to 80 mm and FP = 5 mm because they presented the highest sensitivity and an excellent dynamic range. For this test, it can be observed in Figure 12 a dip shift where the sensitivity obtained was —6.4157 nm/mm, the linear coefficient was 0.9867, and there was a dynamic range of 7.0 mm. If we compare the results obtained in this test with those of FP = 5 mm, there was a sensitivity increase of approximately 7.32 times. The comparison of the results obtained can be observed in Table 2.

## 5. Conclusions

In conclusion, we proposed an experimental investigation of the simple mechanism based on the traditional MMI structure and additive manufacturing for diameter monitoring. We developed two experimental setups for this investigation. The pieces fabrication and the proposed sensor are simple and repeatable. This study demonstrates that it is possible to monitor the diameter using the sensor based on the SMS superimposed on the 3D printer piece proposed in this work. The first results present good sensitivities values, excellent coefficient of determination, and dynamic ranges with several possibilities for practical applications such as in pipelines and tree trunk growth. Based on the initial results shown above, it was possible to develop a new setup without increasing piece and sensor production complexity. The study shows that the new design can increase the sensitivity (7.32 times) significantly and the dynamic range for the analyzed cases using AM. The material used is a biodegradable thermo-plastic of natural origin, resistant, and bio-compatible. These aspects are essential to monitoring the stem growth of a tree, for example, and it is possible to use this measure for deficit water measurement correlation.

It is also possible to optimize the structure and its characteristics to obtain a greater sensitivity or dynamic range. Future studies need to be conducted to confirm the temperature resistance of the material and its influence. Other tests will be developed with different distances between supports. Finally, using other materials and comparing the results to arrive at an optimal sensor is the objective.

## Figures and Tables

**Figure 1 sensors-22-04560-f001:**
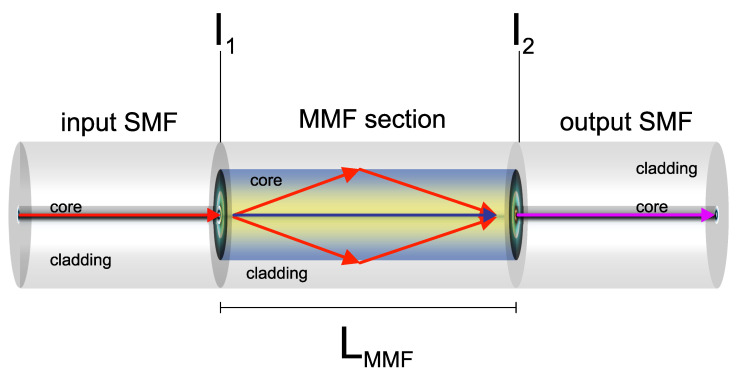
Schematic of the MMI sensor based on the core diameter mismatch technique.

**Figure 2 sensors-22-04560-f002:**
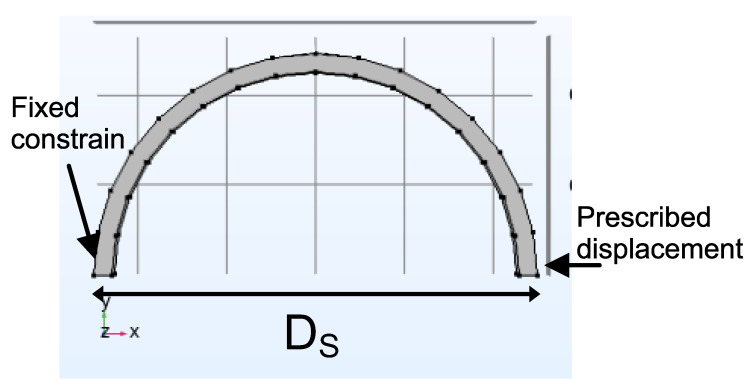
Structural simulation model.

**Figure 3 sensors-22-04560-f003:**
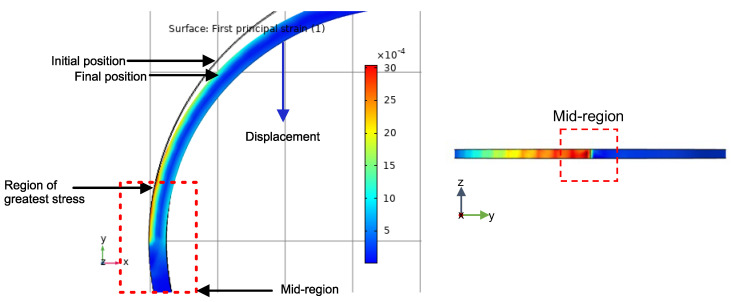
Static structural analysis of the piece without supports based on FEM.

**Figure 4 sensors-22-04560-f004:**
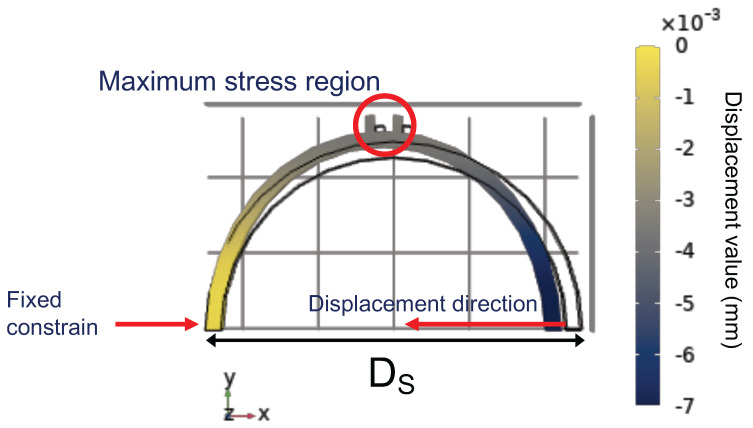
Static structural analysis, based on FEM, of the piece with supports.

**Figure 5 sensors-22-04560-f005:**
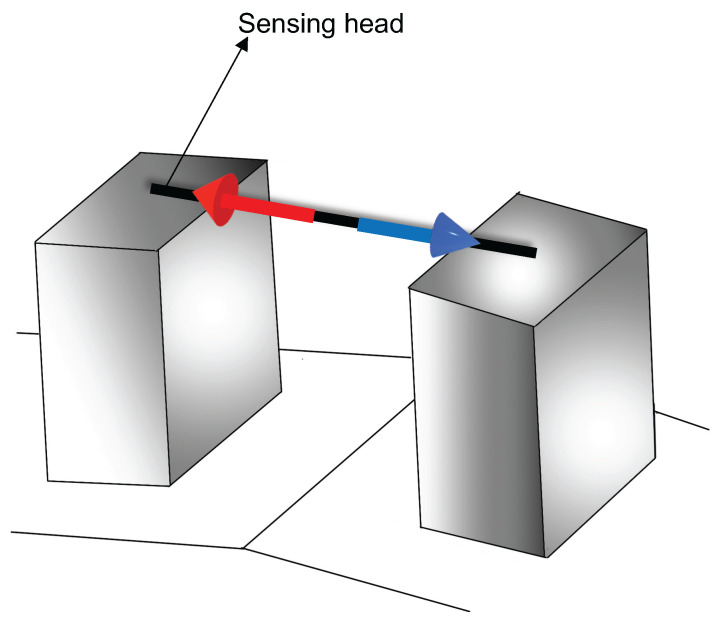
Direction of principal stress.

**Figure 6 sensors-22-04560-f006:**
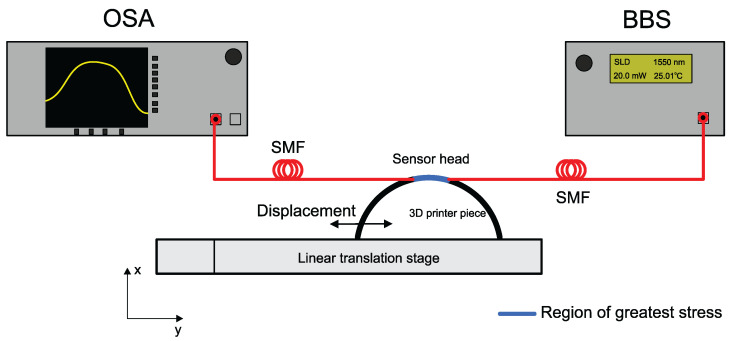
Schematic of the experimental setup for the diameter analysis that demonstrates the diameter reduction using a linear translation stage.

**Figure 7 sensors-22-04560-f007:**
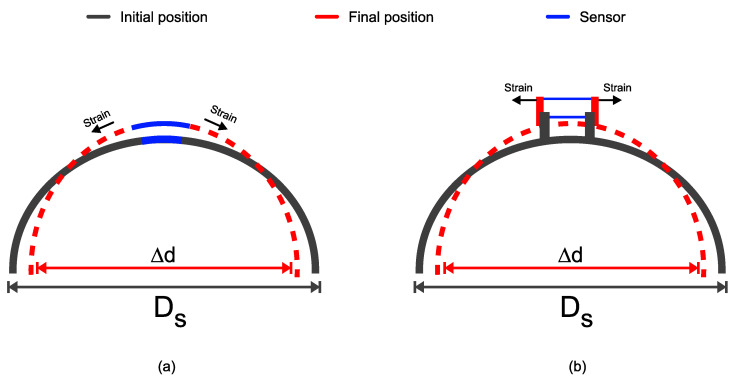
Mechanical operation (**a**) piece without supports; (**b**) piece with supports.

**Figure 8 sensors-22-04560-f008:**
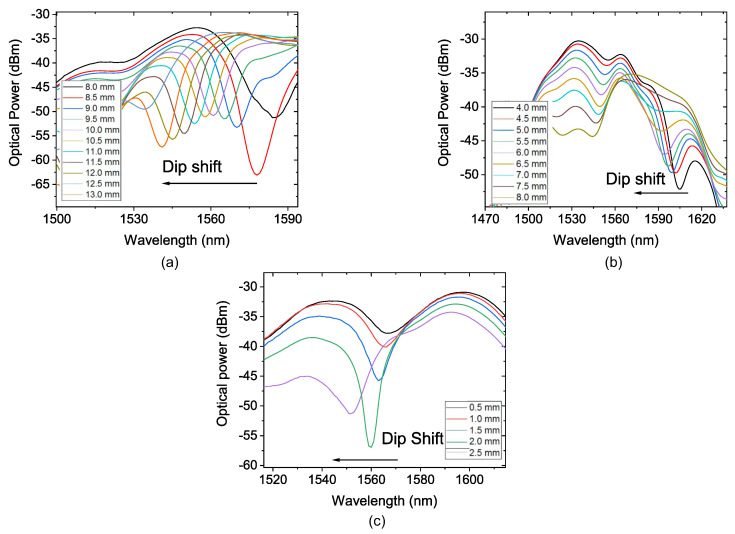
Transmission spectrum of the sensor with D_S_ = 80 mm and (**a**) the sensor with FP = 5 mm; (**b**) the sensor with FP = 10 mm; (**c**) the sensor with FP = 15 mm.

**Figure 9 sensors-22-04560-f009:**
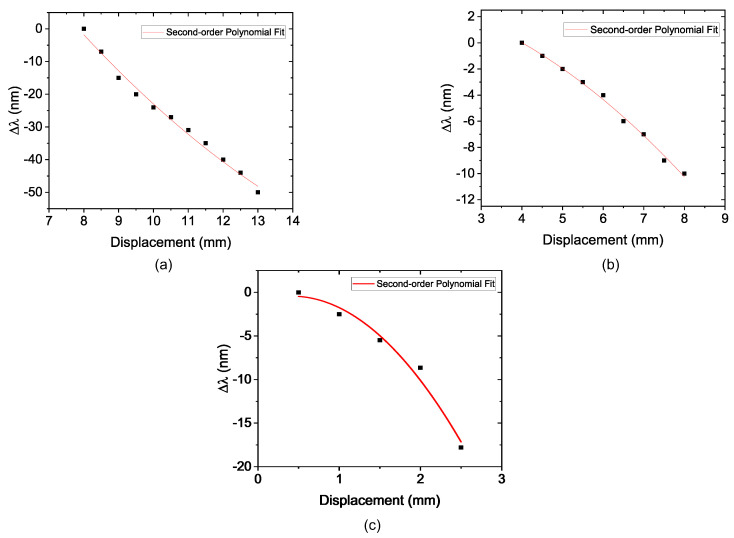
Wavelength shift as a function of displacement with D_S_ = 80 mm and (**a**) the sensor with FP = 5 mm; (**b**) the sensor with FP = 10 mm; (**c**) the sensor with FP = 15 mm.

**Figure 10 sensors-22-04560-f010:**
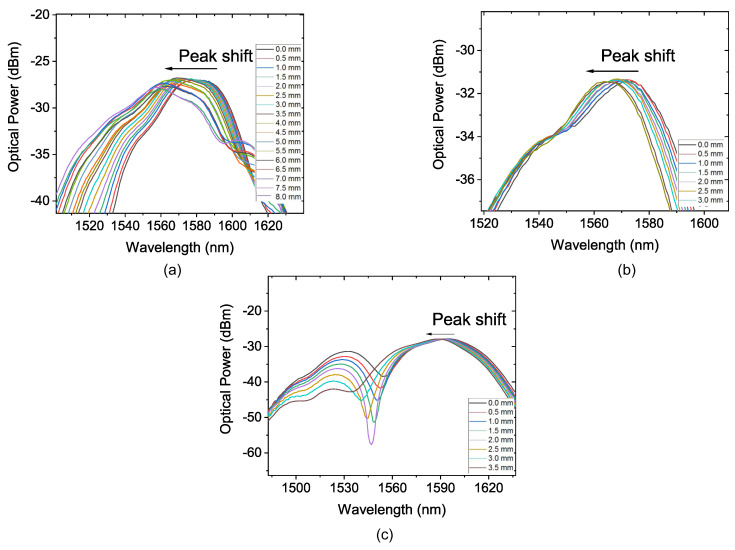
Transmission spectrum of the sensor submitted to diameter variation with D_S_ = 110 mm. (**a**) the sensor with FP = 5 mm; (**b**) the sensor with FP = 10 mm; (**c**) the sensor with FP = 15 mm.

**Figure 11 sensors-22-04560-f011:**
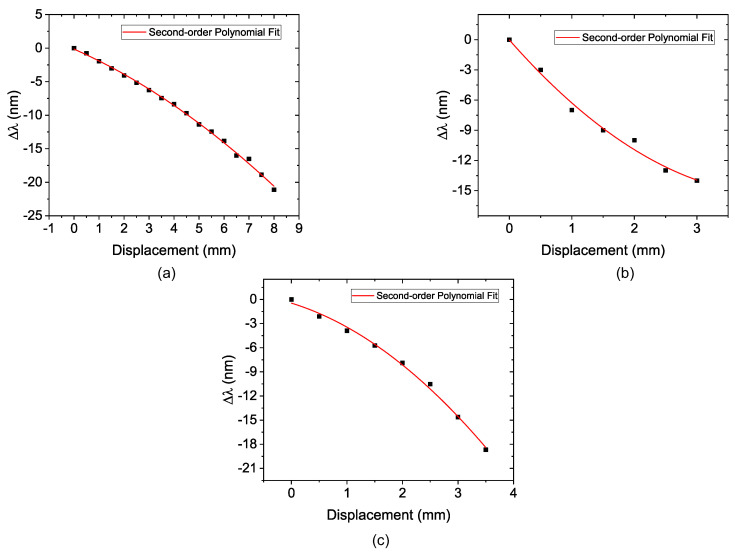
Wavelength shift as a function of displacement with D_S_ = 110 mm and (**a**) the sensor with FP = 5 mm; (**b**) the sensor with FP = 10 mm; (**c**) the sensor with FP = 15 mm.

**Figure 12 sensors-22-04560-f012:**
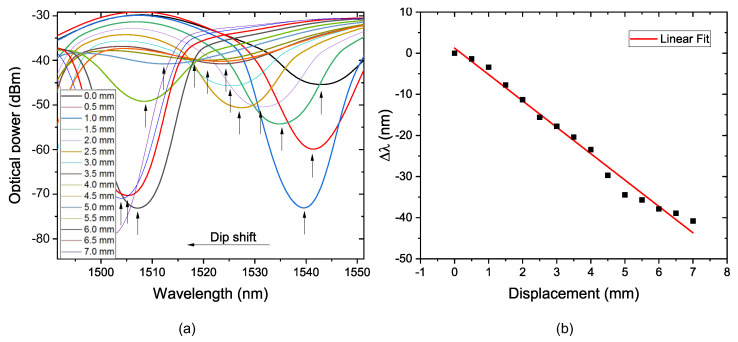
(**a**) Transmission spectrum of the sensor with supports; (**b**) linear fit of wavelength shift as a function of displacement.

**Table 1 sensors-22-04560-t001:** Material properties used in the theoretical static investigation.

Structure	Material	Young’s Modulus (GPa)	Poisson Ratio	Density (kg/m${}^{3}$)
Optical sensor	Silica	66.3	0.15	2.7 × 10${}^{-6}$
Adhesive	Cyanoacrylate	1.26	0.36	1.07 × 10${}^{-3}$
Piece	PLA	3.9	0.33	1.24 × 10${}^{-3}$

**Table 2 sensors-22-04560-t002:** Comparison of the results obtained in this work.

D_S_ (mm)	FP (mm)	Sensitivity (nm/mm)	Dynamic Range (mm)	$R^{2}$
80	5	−0.876	5	0.9909
80	10	−0.3892	4	0.9954
80	15	−0.768	2	0.9811
110	5	−0.22	8	0.9979
110	10	−0.2284	3	0.9888
110	15	−0.691	3.5	0.9892
80 with support	5	−6.4157	7.0	0.9867

## Data Availability

Not applicable.

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
