# Peer review of "A Simple Optical Sensor Based on Multimodal Interference Superimposed on Additive Manufacturing for Diameter Measurement†"

_sensors, 2022, doi:10.3390/s22124560_

Round 1
Reviewer 1 Report
The manuscript presented a SMS sensor structure which comminated a cylindrical piece developed using a 3D printer and a Mach-Zehnder interferometer. The work has been demonstrated be simple and practicability. But there are still some issues to be modified.
1. In Fig.3, why did not the author give the full FEM image? Where was the “mid-region” in Line 152 page 5 for Fig.3
2. The authors should indicate the location of the parameter “Ds” in all related Figures. In Fig.7, it should be “Ds” not “ds”.
3. How about the sensor’s performance when the load is a compression?
4. It is not clear what are the means of the red arrows and the color bar in Fig.5.
5. In Fig.10, the authors should indicate the variations of the diameter for all the lines.
6. In Fig.11, the fitting results should be given, and how did the authors define the sensitivity by the second-order polynomial fitting result?
Author Response
Dear Reviewer 1,
Please see the attachment.
Kind regards,
Victor Cardoso

Reviewer 2 Report
In this work it was presented a displacement sensor based on an interferometric sensor. Here the sensing element was fixed over a piece fabricated with a 3D printer. In general the work is interesting and is supported with a set of experimental measurements. Therefore I consider that the work can be considered for publication after major changes are carried out in order to clarify some points of the manuscript
Some points that must be addressed are:
11- In page 4, lines 128-130, it is stated ‘From the derivation of Equation 4, it is possible to determine the transmission spectrum of the SMS sensor from the refractive index of the core, the diameter of the core, and the length of the MMF [49]’. Here please check the meaning of this sentence because it is not clear how by derivation of eq. (4) the transmission spectrum can be determined.
22- In page 5, line 148, it is written ‘Considering, for example, the use in specimens from the Amazon,…’, here consider to write it as something as ‘Considering, for example that, the sensor will be used at the Amazon,…’.
33- In page 6, lines 157-160, is a bit confusing the sentence ‘The sensor behavior output power intensity is a function of the curvature radius, and it is directly associated with the power distribution in the MMF section, where modes are excited and they interfere as a function of bending’. I suggest to rewrite the sentence to clarify it and also check if just the power intensity is affected by the curvature or if also the transmitted intensity distribution (spectrum), since later the sensor output is determined by means of a relationship between the wavelength shift of a dip and the displacement.
44- I consider that authors must enhance in general the description of the fabricated 3D piece and how the experiment was implemented, because right now is quite difficult to follow and figures do not help much. For instance in page 4 it is commented that ‘as depicted in Figure 4. It is possible to observe that the supports show a shift between them. Due to it, when the 3D printer piece is submitted to diameter size (DS) modification, the sensor that is fixed over the piece is tensioned’. However, in Fig. 4 it is difficult to observe the described by authors. Also I suggest to put some annotations within the figure in order to understand what it is presented.
55- In the manuscript are described two experimental cases, however it is difficult to understand the differences between the 2 cases, again Fig. 7 is not quite descriptive. Please try to explain what are the main differences, how the sensor element is attached etc., in order to get a better picture of how the experiments were implemented.
66- Previous point is related to the end of section 3 where it is described the two cases but it needs to be clarified.
77- In figure 8 and 10 there are presented transmission spectra for different sensors. Please for each plot add legends to identify the conditions for which each spectra were taken. For instance according to the manuscript text spectra at Fig. 8 are for different curvatures, however the corresponding curvatures do not appears in the any plots.
88- please clarify or explain how the curvature was related to the displacement, because in figure 10 there are presented the wavelength shifts as a function of displacement, while it was stated that the spectra of Fig. 8 show spectra for different curvatures. Similar issue with the case of figure 11
99- One issue that it is commented in section 4 is the measurement or dynamic range achieved with the different sensors in mm scale. Please discuss how this value was determined, since I consider it is quite important parameter of the sensor.
110- Regarding with the last part of section 4, where the FFT magnitude is used to explain the curvature and strain effects over the sensor head, I would like to comment that it is really difficult to observe some differences between the spectra, and based on these appreciate the curvature and strain effects. So I would like to ask to enhance the explanation of this section and if possible provide other figures where effects can be better appreciated or even consider to eliminate this part that from my point of view, at this moment, do not provide conclusive information.
111- As minor corrections, in some parts of the manuscript mm and nm units are written in italics and in other as normal style. Please homogenize it, and I suggest to use normal style since are units.
Author Response
Dear Reviewer 2,
Please see the attachment.
Kind regards,
Victor Cardoso

Reviewer 3 Report
This paper calls: «A simple optical sensor based on Multimodal Interference superimposed on additive manufacturing for diameter measurement» and concerned of diameter measurement system by bending of conventional SMS fiber system. The advantages of work are interesting topic of research and long references list.
1.What about bending birefringence of bending part?
2. Authors have any ideas how to build distributed measurement system based on this principle of operation?
Author Response
Dear Reviewer 3,
Please see the attachment.
Kind regards,
Victor Cardoso

Round 2
Reviewer 1 Report
TThe manuscript can be accepted in present form.
Reviewer 2 Report
Authors have attended the reviewer comments and have enhanced their manuscript. Therefore I would like to suggest to publish the manuscript in its present form.